# Is Spheroid a Relevant Model to Address Fibrogenesis in Keloid Research?

**DOI:** 10.3390/biomedicines11092350

**Published:** 2023-08-23

**Authors:** Zélie Dirand, Marion Tissot, Brice Chatelain, Céline Viennet, Gwenaël Rolin

**Affiliations:** 1Université de Franche-Comté, Sciences Médicales et Pharmaceutiques, EFS, INSERM, UMR RIGHT, 25000 Besançon, France; zelie.dirand02@edu.univ-fcomte.fr (Z.D.);; 2Service de Chirurgie Maxillo-Faciale, Stomatologie et Odontologie Hospitalière, CHU Besançon, 25000 Besançon, France; 3Université de Franche-Comté, Sciences Médicales et Pharmaceutiques, CHU Besançon EFS, INSERM, UMR RIGHT, 25000 Besançon, France

**Keywords:** spheroid, keloid fibroblast, fibrosis, ECM, α-SMA, TGF-β1

## Abstract

Keloid refers to a fibro-proliferative disorder characterized by an accumulation of extracellular matrix at the dermis level, overgrowing beyond the initial wound and forming tumor-like nodule areas. The absence of treatment for keloid is clearly related to limited knowledge about keloid etiology. In vitro, keloids were classically studied through fibroblasts monolayer culture, far from keloid in vivo complexity. Today, cell aggregates cultured as 3D spheroid have gained in popularity as new tools to mimic tissue in vitro. However, no previously published works on spheroids have specifically focused on keloids yet. Thus, we hypothesized that spheroids made of keloid fibroblasts (KFs) could be used to model fibrogenesis in vitro. Our objective was to qualify spheroids made from KFs and cultured in a basal or pro-fibrotic environment (+TGF-β1). As major parameters for fibrogenesis assessment, we evaluated apoptosis, myofibroblast differentiation and response to TGF-β1, extracellular matrix (ECM) synthesis, and ECM-related genes regulation in KFs spheroids. We surprisingly observed that fibrogenic features of KFs are strongly downregulated when cells are cultured in 3D. In conclusion, we believe that spheroid is not the most appropriate model to address fibrogenesis in keloid, but it constitutes an efficient model to study the deactivation of fibrotic cells.

## 1. Introduction

Keloid refers to a fibro-proliferative disorder characterized by an accumulation of extracellular matrix (ECM) components at the dermis level, overgrowing beyond the initial wound and forming tumor-like nodule areas [1]. Whatever the trauma, keloids always start from skin lesions and are the consequence of a dysregulated healing process. Experts now consider keloids as a chronic inflammatory disease [2] that shares close similarities with cancer [3]. Clinically, keloids are benign; however, they seriously impair patients’ quality of life, especially when they are located on the face and joints. Moreover, keloids can cause itching, pain, and discomfort in patients. Unfortunately, a treatment for keloids is yet to be uncovered [4], and the lack of an efficient therapy is clearly related to limited knowledge about keloid etiology, despite the increasing number of publications on the subject. 

Molecular mechanisms in keloids still need to be deciphered. This challenge remains difficult because of the lack of relevant animal models to efficiently address keloid fibrogenesis [5]. However, various models have been developed to study keloid disease, including in silico, in vitro, ex vivo, and in vivo models, as reviewed by Lebeko et al. [6]. All these tools became essential in keloid research in order to explore keloid fibroblast biology, screen anti-fibrotic drugs [7], and discover new biomarkers [8]. 

Whatever the experimental model, numerous research were carried out through the prism of fibroblast as the major effector of ECM deposition [9]. In keloids, fibroblasts are present in high numbers compared to normal tissues [10]. Keloid fibroblasts (KFs) express higher rates of α-Smooth Muscle Actin (α-SMA) [11] and collagen [12] than normal dermal fibroblasts (NDFs). KFs also have a higher proliferation rate [13] and are able to develop higher retraction forces [14,15]. In addition, KFs are more sensitive to their biological microenvironment, as they have more TGF-β (TGFβRI and TGFβRII, the two first sensors of TGF-β [16]) and PDGF receptors than NDFs [17,18,19]. KFs are more responsive to growth factors that upregulate myofibroblast differentiation and over-amplify collagen and ECM synthesis and deposition [20]. 

KFs cultured in 2D do not fully recapitulate the in vivo quasi-neoplastic profile, architecture, cell–cell, and cell-matrix interactions observed in keloid tissue. Recent data published from single-cell investigations [8,21] highlight the close communication network between fibroblasts, keloid-associated immune cells (i.e., macrophages and dendritic cells), and keloid-associated stem cells [22], which can all modulate fibrogenesis of KFs. Gathering all these cell types into the same in vitro model could be technically difficult. 

In response to these limitations, researchers proposed several 3D models to propose an intermediate complexity between monolayers and keloid tissue; for example, KFs embedded in 3D collagen gels or dermo-epidermal reconstructed keloid. Closer to in vivo, keloid explants obtained from surgical procedures can also be maintained in culture and used as an in vitro platform for experimentation. In proper conditions, explants retain the main characteristics of fibrotic tissue (i.e., TGF-β1 expression and collagen content) [23,24,25]. These models can also be customized thanks to the addition of cells, ECM, and biological factors around the sample in order to create an in vivo-like microenvironment [6]. However, the main limitation of this model is the required complex logistic to have frequent access to fresh tissue from surgery in the right regulatory environment. 

Recently, spheroids have emerged as new tools for tissue engineering and cancer research to mimic organs or diseases as a replacement for animal models [26]. A spheroid is a 3D aggregate of cells which spontaneously forms when attachment to a substrate is prevented [27]. In spheroids, cells create cell–cell interaction and are able to generate their own ECM micro-environment similar to in vivo conditions. Spheroids are now widely used for treatment screening [27,28,29], namely in cancer research. Spheroids have also already been considered as a tool for research in cutaneous biology [30,31]. But to our knowledge, no previous works have specifically focused on keloid pathology yet. Thus, we hypothesized that spheroids made from keloid fibroblasts could be used to model fibrogenesis in vitro. Indeed, keloid fibroblasts are the main effectors of ECM deposition, and 2D KFs cell culture has been the main tool for understanding keloid physiopathology for a long time. However, monolayer cultures of KFs are very far from the in vivo reality and complexity of keloids.

The objective of our study was to qualify spheroids made from keloid fibroblasts (KFs) in comparison to normal dermal fibroblasts (NDFs). To this aim, we produced KFs and NDFs spheroids and cultured them in a basal or pro-fibrotic environment (with TGF-β1). To fully characterize the 3D fate of our cells, we also classically cultured them in 2D as an ultimate control. As major parameters for fibrogenesis assessment in KFs, we evaluated apoptosis, fibroblast-to-myofibroblast differentiation (α-SMA and CD26 expression) and response to TGF-β1, ECM synthesis, and ECM-related genes regulation. Regarding our results, we surprisingly highlighted that KFs are strongly inactivated when cultured from 2D to 3D and that they lost their sensitivity to TGF-β1 in spheroids. In consequence, α-SMA and collagen expression is reduced to basal level. In conclusion, we believe that while spheroid is not the expected relevant model to address fibrogenesis in keloids, it constitutes an efficient tool to study the deactivation of fibrotic cells and offers new perspectives for keloid research.

## 2. Materials and Methods

### 2.1. Clinical Study Approval

Keloid tissues were obtained from patients undergoing reductive plastic surgery performed at Maxillo-Facial Surgery Department of the University Hospital of Besançon (CHU de Besançon, France). All included patient provided informed consent and the study was conducted in accordance with the ethical standards, namely the Declaration of Helsinki. This work was ethically approved by the French Regulatory Agency (ANSM), ethic committee (CPP Sud-Ouest and Outre-Mer I) and was registered on clinicaltrial.gov as “SCAR WARS” (NCT03312166). Normal skin was obtained from abdominal dermo-lipectomy performed during routine surgical procedure and after informing the patient and obtaining their consent. Sex and age of donors are listed in Table 1.

### 2.2. Human Normal Dermal (NDFs) and Keloid Fibroblasts (KFs) Collection

Normal dermal fibroblasts (NDFs) and keloid fibroblasts (KFs) were, respectively, isolated from abdominal dermolipectomy and earlobe keloid. NDFs and KFs were isolated as previously described [7]. After outgrowth, KFs were subcultured in complete DMEM (5% FCS, 1% PS) and used between the third and eight passage for all experiments.

### 2.3. Monolayer and Spheroid Cultures

In 2D, NDFs and KFs were seeded at 2 × 10^5^ cells/well in 12-well plates. For 3D spheroids formation, NDFs and KFs were seeded at 4.5 × 10^4^ per well in Ultra Low Attachment (ULA) 96-well culture plates (MS-9096UZ Prime Surface^®^ 3D culture, S-Bio). During both 2D and 3D situation, cells were cultured in “control” condition (DMEMc) or in “TGF-β1” condition (DMEMc + 10 ng/mL TGFβ-1 [240-B-002, R&D Systems, Minneapolis, MN, USA]). For both monolayers and spheroids, half of the medium was renewed after 4 days of culture, and cells were harvested for analysis after spheroid maturation period (7 days). The size of spheroids was monitored every 24 h for 7 days using IncuCyteS3^TM^ real-time microscope. 

### 2.4. Histological and Immuno-Histological Characterization of Spheroids

Spheroids were collected after 7 days of culture, washed with PBS 1X, and fixed in 4% paraformaldehyde (Sigma Aldrich, Saint-Louis, MO, USA). After fixation, spheroids were embedded in TissuTek^®^ O.C.T (Sakura Finetek, Tokyo, Japan) and frozen at −80 °C. HES staining was first used to characterize spheroid architecture. Moreover, 7 µm spheroid sections from TissuTek^®^ were washed in distilled water, and then incubated in Harris hematoxylin solution (3 min, RT). Sections were washed with tap water, incubated with eosin (1 min, RT), washed again with distillated water, and incubated in 1% acetic water. Sections were dehydrated in 100% ethanol (10 min, RT) before staining in saffron solution (5 min, RT). Sections were mounted between a glass slide and coverslip before microscopic imaging using Axioskop40 epifluorescence microscope (Carl Zeiss, Oberkochen, Germany). For immunostaining, spheroid sections were manipulated as follows: 7 µm spheroid slice were first permeabilized in 0.1% Triton X100. Then, blocking solution (PBS, 3% BSA, 10% sheep serum) was used to saturate aspecific binding sites. Appropriate dilutions of primary antibodies targeting αSMA (A2547, Sigma Aldrich, Saint-Louis, MO, USA), CD26 (OTI11D7, OriGene, Rockville, MD, USA), and TGFβRII (PA5-35076, Invitrogen, Carlsbad, CA, USA) were added on glass slides before incubation (overnight, 4 °C in a wet chamber). Then, sections were incubated (1 h, RT) with appropriate secondary antibodies (F8521 and AP307R, Sigma Aldrich, Saint-Louis, MO, USA). Nuclei were counterstained with DAPI (15 min, RT). A negative control was obtained by omitting primary antibodies. Images were obtained using a LSM800 confocal microscope (Carl Zeiss, Oberkochen, Germany). Fluorescence quantification was performed using ImageJ. Results are expressed as a relative fluorescence intensity (RFI). RFI was calculated as protein fluorescence (green) normalized by DAPI intensity (blue). Raw data were obtained from independent experimentation performed with three different primary cell lines of NDFs and three primary cells lines of KFs (*n* = 3 per cell line and condition). The fluorescence quantification was performed on at least 7 spheroids slides per culture conditions and per cell line.

### 2.5. Terminal Deoxynucleotidyl Trasferas dUTP Nick End Labeling (TUNEL) Assay

Spheroid sections were produced as described in Section 2.4, and TUNEL assay was performed using One-Step TUNNEL In Situ Apoptosis kit (Elabscience, Houston, TX, USA), following manufacturer’s instructions. Briefly, cells were permeabilized with proteinase K solution (10 min, 37 °C). Labeling solution containing TdT enzyme was then added on slides and incubated 1 h at 37 °C. Then, nuclei were counterstained with DAPI, washed with PBS, and mounted between slide and coverslip. Stained samples were imaged with a LSM900 laser scanning confocal microscope (Carl Zeiss, Oberkochen, Germany). Positives cells (red) were counted using ImageJ, and results were expressed as a number of apoptotic cells per mm^2^. Raw data were obtained from independent experimentation performed with two different primary cell lines of NDFs and three primary cells lines of KFs (*n* = 2 per cell line and condition). The count of TUNEL positive cells was determined on a minimum of 7 spheroids per culture conditions and per cell line. 

### 2.6. RT-qPCR

After treatment, cells from 2D and 3D cultures were harvested and lysed in RLT buffer supplemented with 4% dithiothréitol. RNA extraction was performed using RNeasy mini kit (Qiagen, Venlo, The Netherlands), and RNAs were reverse-transcribed using High capacity RNA to cDNA kit (ThermoFisher, Waltham, MA, USA) according to manufacturer’s instructions. Then, specific TaqMan probes from Thermo Fisher Scientific (TaqMan Gene Expression Assays-*ACTA2*: Hs00426835_g1, *DPP4*: Hs00897405_g1, *TGFbRII*: Hs00234253_m1, *COL1A1*: Hs00164004_m1, *COL3A1*: Hs00164103_m1 and *FN1*: Hs01549976_m1) were used for qPCR amplification. All samples were run in duplicate. A normalization of the RNA level was performed against the GAPDH gene. Data were analyzed using the ΔCT technique [32] and are represented as a relative expression compared to the monolayer control condition. Raw data were obtained from independent experimentation performed with three different primary cell lines of NDFs and three primary cells lines of KFs (*n* = 1 per cell line and condition).

### 2.7. α-Smooth Muscle Actin and Fibronectin Quantification

After 7 days, concentration of α-SMA and fibronectin in 2D and 3D culture were quantified using Human α-SMA ELISA (ab240678, Abcam, Cambridge, United Kingdom) and Human fibronectin ELISA (ab219046, Abcam, Cambridge, United Kingdom) according to manufacturer instructions. Data were normalized to the number of total proteins in samples, which was determined using Pierce BCA reaction. Raw data were obtained from independent experimentation performed with three different primary cell lines of NDFs and KFs (*n* = 3 per cell line and condition).

### 2.8. Statistical Analysis

Results are expressed as mean ± SD. Statistical analyses were performed using two-way analysis of variance (ANOVA). All analyses were performed using GraphPad Prism 9 software. Differences were considered as statistically significant * for *p* < 0.05; ** for *p* < 0.01; *** for *p* < 0.001; **** for *p* < 0.0001.

## 3. Results

### 3.1. KFs and NDFs Diameter Equally Evolve Overtime Independently from TGF-β1 Activation

The evolution of KFs and NDFs spheroids was followed over time, and their diameters were measured daily in each culture condition (Figure 1A). No difference was observed between KFs and NDFs neither in control nor in profibrotic condition (TGF-β1). In each spheroid batch, we observed a large compaction phase during the first 12 h (from 10^7^ to 2 × 10^6^ µm^2^), followed by the stabilization of the phenomenon until the third day. Then, spheroid diameter reached a plateau (2 × 10^5^ µm^2^) at the end of the follow-up period (7 days). Similarly, no differences were observed in spheroids’ architecture, whatever the fibroblast origin or the culture condition, as shown by HES staining in Figure 1B.

### 3.2. TGF-β1 Reduced Apoptotic Cells Rate in KFs Spheroids More Than in NDFs Ones

We performed a TUNEL assay to label apoptotic cells in spheroid section and assess cell viability in 3D structure after 7 days of maturation (Figure 2A). Apoptotic cells (red) were counted from picture series and cell number per surface unit was quantified and presented in (Figure 2B). Results show that the number of cells undergoing apoptosis are over numbered in NDFs compared to KFs spheroids cultured in control conditions. Interestingly, in both NDFs and KFs, TGF-β1 reduces the quantity of apoptotic cells in 3D.

### 3.3. TGF-β1-Induced α-SMA Expression Discontinues in KFs Spheroids

α-Smooth Muscle Actin (α-SMA) was investigated as the main marker of fibroblast-to-myofibroblast transition. We analyzed α-SMA regulation in KFs and NDFs cultured in 2D and 3D by RT-qPCR, ELISA, and immunostaining (Figure 3). First, we observed that α-SMA protein level was strongly increased in KFs monolayer in response to TGF-β1 activation (6.34 µg α-SMA/g total proteins vs. 44.70, *** *p* = 0.0004) compared to NDFs (4.78 vs. 6.45, *p* > 0.9999). These observations were also confirmed at the mRNA level (Figure 3B). Switching from 2D to 3D, the TGF-β1 induction of α-SMA expression discontinued in KFs spheroids compared to monolayer culture. In spheroids, previous results were confirmed by immunofluorescence (Figure 3C,D), which showed that α-SMA expression is equivalent in spheroids, whatever the origin of fibroblasts (normal or keloid) and the nature of the treatment (control or TGF-β1).

### 3.4. Three-Dimensional Culture and TGF-B1 Activation Converge to Downregulate CD26 and TGFβRII Expression

To evaluate the fibrogenic level of fibroblasts and their capacity to respond to TGFβ-1 activation, we investigated mRNA (Figure 4A,B) and protein (Figure 4C,E) expressions of *TGFβRII* (TGFβRII) and *DPP4* (CD26) in confluent monolayer or 3D spheroids. We observed that TGFβRII transcription was significantly more important in NDFs confluent monolayers than in KFs ones, in both non-fibrotic and fibrotic conditions (Figure 4A). Interestingly, moving from 2D to 3D culture leads to the decrease in TGFβRII transcription profile both in NDFs, in control or treated conditions. Figure 4B presents fold change in *DDP4* expression that is not impacted by TGF-β1 treatment or spheroids culture in NDFs. However, we showed that *DDP4* mRNA expression is downregulated in KFs confluent monolayers compared to NDFs. Both in 2D and in 3D, this expression is even more restricted by TGF-β1 treatment.

Complementary results obtained by immunostaining in spheroids are presented in Figure 4C and related semi-quantification in Figure 4D,E. We can see that that the quantity of CD26 (green) and TGFβRII (red) significantly decrease in TGF-β1-treated KFs spheroids compared to control (0.62 vs. 1.01 for CD26, * *p* = 0.0270 and 0.55 vs. 1.00 for TGFβRII, *p* = 0.1008), while there is no impact of the culture conditions on NDFs regarding the expression of both proteins.

### 3.5. Overexpression of ECM Related Genes Discontinues When KFs Are Cultured from 2D to 3D

At the protein level, fibronectin expression (Figure 5A) is upregulated by TGF-β1 both in KFs and NDFs monolayers. In 3D, fibroblast sensitivity to TGF-β1 remains in both cell lines. However, protein quantity decreases in KFs spheroid compared to 2D culture (0.035 vs. 0.219 in Control, *p* = 0.9914 and 0.312 vs. 1.434 in TGF-β1, ** *p* = 0.0037). The same tendency was observed in NDFs spheroids compared to monolayers. The same tendencies were observed at mRNA level, as shown in Figure 5B. Our results showed that *COL1A1* transcription is upregulated by TGF-β1 in 2D both in NDFs and KFs. When cells are cultured in 3D, this effect is interrupted, as seen in Figure 5C. *COL3A1* mRNA transcription was also upregulated by TGF-B1 treatment in NDFs cultured either in monolayers or in 3D spheroids (Figure 5D). Surprisingly, we observed that culturing keloid fibroblasts in 3D leads to an 11-times increase in fold ratio (** *p* = 0.0057), and that TGF-β1 effect is reversed compared to monolayer culture. We calculated COL1A1/COL3A1 ratio as a fibrogenesis marker in fibroblasts (Figure 5E). We showed that this ratio is strongly increased in KFs compared to NDFs in 2D culture (3.222 vs. 1.000 in Control, *p* = 0.1092). This upregulation is even higher in TGF-β1-treated keloid fibroblasts (5.177 vs. 1.092 in TGF-β1, *** *p* = 0.0001). However, COL1A1/COL3A1 is strongly reduce in KFs 3D culture compared to monolayers, in both control and treatment condition (0.320 vs. 3.222 in Control, *** *p* = 0.0001 and 1.023 vs. 5.177, **** *p* < 0.0001).

## 4. Discussion

Keloids are a fibro-proliferative skin disorder which can seriously affect a patient’s quality of life. The lack of highly efficient treatment is strongly related to the absence of a reference model for the development of novel therapies. Nevertheless, as recently reviewed by Limandjaja et al. and Supp et al., numerous in silico, in vivo, and in vitro models have recently been proposed to investigate keloid [5,33]. Among them, full-skin equivalent or explant (also called organoids) are very useful tools as an in vitro platform for experimentations and anti-fibrotic drug screening [34,35,36]. However, these two models aim to reproduce or maintain keloid tissue in vitro and mimic an already established fibrotic tissue. Of course, explants retain the main characteristics of fibrotic tissue (i.e., TGF-β1 expression and collagen content) [23,24,25], but they do not address dynamic evolution of keloid after wounding. In consequence, we hypothesized that spheroids made from keloid fibroblasts (KFs) could be relevant for fibrogenesis research. To this aim, we qualified spheroids made from keloid fibroblasts (KFs) and cultured in a pro-fibrotic micro-environment (TGF-β1) in comparison to normal dermal fibroblasts (NDFs).

Cancer spheroids usually overgrow over time as a consequence of the pathological cell phenotype [37,38,39]. In our study, we did not observe such overgrowth, but we measured a decrease in the spheroid size during the culture period. While keloid is described as a pseudo-cancer pathology, keloid fibroblasts do not share pathological specificities with cancer cells leading to continuous over-proliferation in vitro. Spheroid surface evolution was linked to self-contraction of the 3D construct, mediated by cell–cell interaction and contractile capacity [40], as previously shown in normal dermal fibroblasts by another team [31,41]. Our results showed that spheroid compaction was similar with NDFs and KFs either cultured in basal or profibrotic conditions. After spheroid maturation (plateau of contraction), NDFs showed a higher apoptosis rate in spheroids than KFs. In addition, TGF-β1 treatment decreased even more in both NDFs and KFs. These 3D observations are in accordance with in vivo and in vitro descriptions of the KFs refractory status to apoptosis compared to normal cells [13,42,43,44]. Such specificity can be related to autocrine TGF-β1 stimulation [43] and upregulation of the NF-kB pathway [42] in KFs. Due to spheroid architecture, we can also assume that hypoxia could modulate apoptosis in such 3D culture. Indeed, Lei et al. [45] previously mentioned that hypoxia could decrease apoptosis and mediate proliferation in KFs but not in NDFs. As hypoxia in KFs is also associated with an increase in collagen synthesis [46], the role of oxygen starvation on KFs in such specific 3D spheroids could further be addressed in our model.

Fibrogenic markers of fibroblasts were studied by measuring fibroblast-to-myofibroblast transition, ECM deposition, TGFβRII, and CD26 expression as well. CD26 has been lately proposed as a novel fibroblast activation marker [47,48]. Recent studies highlighted that CD26^+^ fibroblasts expressed higher collagen rate, fibronectin and TGF-β1 compared to CD26^−^ cells [47,49]. In association with FAP (Fibroblast Activation Protein), CD26 expression would mediate ECM synthesis toward the TGF-β/Smad pathway [48]. We compared mRNA levels and/or protein expression between 2D and 3D cultures treated with TGF-β1 or not. In 2D, our results confirmed previous data showing that KFs are more sensitive to TGF-β1 in 2D than NDFs, regarding α-SMA expression and ECM deposition [7,18,50]. TGFβRII and CD26 level expression were lower in KFs than in NDFs. Moreover, TGF-B1 reinforced this effect. Our observation on TGFβRII confirmed those of Smith et al. [51]. Despite low TGFβRII expression, KFs remained highly sensitive in 2D to TGF-β1 regarding α-SMA expression. Concerning CD26 (*DPP4*) expression, our results are in accordance with those of Chen et al. [52], who showed that keloid fibroblasts were DPP4^low^/TGFβ-1^high^ compared with DPP4^high^/TGFβ-1^low^ fibroblasts in normal skin tissue. In 3D spheroids, basal levels of α-SMA and fibronectin expression, as well as *COL1A1/COL3A1* ratio, were decreased in KFs compared to monolayers. In our model, TGF-β1-treated KFs lost their ability to differentiate into myofibroblasts. Our data converge to those of Granato et al. [31] and Kunz-Schughart et al. [53], who cultured NDFs in spheroid models and observed the same deactivation effect on normal cells. We strongly believe that this fibrogenic deactivation effect is connected with growing keloid fibroblasts as multicellular aggregates. Indeed, in our model, keloid fibroblasts do not have any surface to adhere to apart from each other. In spheroids, the mechanical component (high rigidity of plastic surface) found in 2D cultures has disappeared, while matrix stiffness is strongly mandatory for keloid fibroblast activation [11], independently of TGF-β1.

Our first objective was to produce a spheroid model to explore fibrogenesis in keloid fibroblasts. Beyond our first expectations, we demonstrated that such a model is advantageous and an efficient tool to study the deactivation of fibrotic cells and offer new perspectives for keloid research. However, we identified some limitations that should be overcome in further studies. Particularly, we propose to replicate our investigation with pre-activated fibroblasts (with TGF-β1) to see if the deactivation effect is still active on pre-differentiated cells. We also propose to follow spheroid behavior when KFs are co-cultured with immune cells (i.e., macrophages) in order to address cell–cell interaction in this context. Because our model lacks a surrounding matrix, we could also further study cell outgrowth and invasion in KFs spheroids surrounded by a specific ECM micro-environment (collagen, fibronectin, or Matrigel^®^).

## 5. Conclusions

At the beginning of our work, our first hypothesis was to succeed in producing a fibrotic model of keloids shaped as spheroids. Our study demonstrated that spheroids from human keloid fibroblasts can be generated and maintained in culture but trigger a deactivation effect of fibrotic cells. Fibrogenic features of KFs were strongly downregulated when cells were cultured in such 3D structures. Even if our spheroid is not the expected relevant model to address fibrogenesis upregulation, our work highlights new aspects of turnover in keloid cells. Keloid spheroids constitute an efficient tool for studying the deactivation of fibrotic cells and offer new perspectives for keloid research.

## Figures and Tables

**Figure 1 biomedicines-11-02350-f001:**
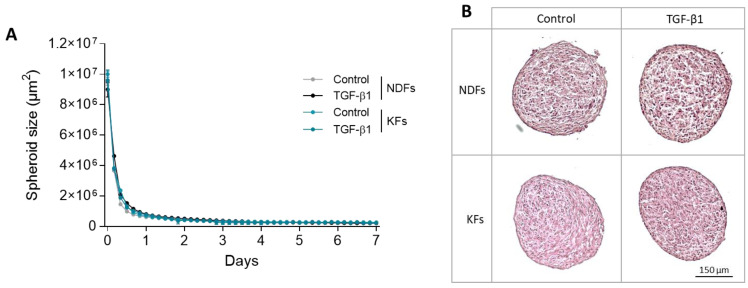
KFs and NDFs area equally evolve over time independently from TGF-β1 activation. To generate spheroids, NDFs or KFs were seeded in ULA culture plates +/− TGF-β1. (**A**) Diameters were followed over time using IncuCyteS3 microscope. (**B**) After maturation (7 days), spheroids slices were stained with H&E for architecture study. Results are expressed as mean ± SD. Statistical analyses were performed using two-way ANOVA (*n*= 16 spheroids per condition).

**Figure 2 biomedicines-11-02350-f002:**
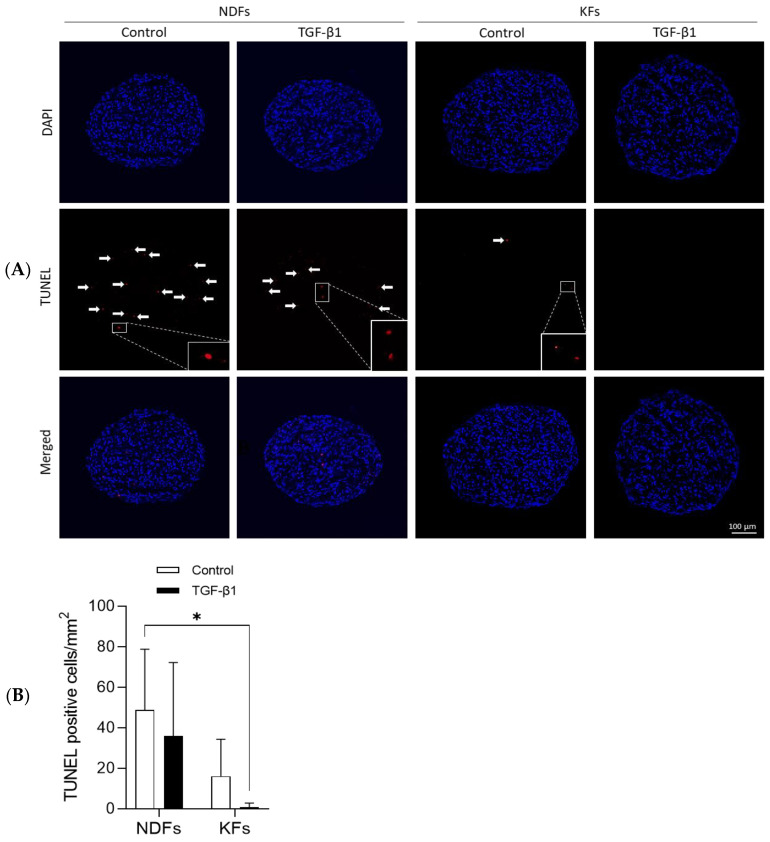
TGF-β1 reduced apoptotic cells rate in KFs spheroids more than in NDFs ones. (**A**) After 7 days, NDFs and KFs spheroids (either culture in control medium or with 10 ng/mL TGF-β1) were prepared for TUNEL staining (7 µm spheroid section). Images from confocal microscope show total nuclei in blue and those from apoptotic cells in red (highlighted by the white arrows). (**B**) TUNEL positive cells were counted on spheroid sections. Results are represented as a number of positive cells/mm^2^. A minimum of five slides were used for each spheroid [*n* = 4 spheroid per condition]. Statistical analyses were performed using two-way ANOVA * for *p* < 0.05.

**Figure 3 biomedicines-11-02350-f003:**
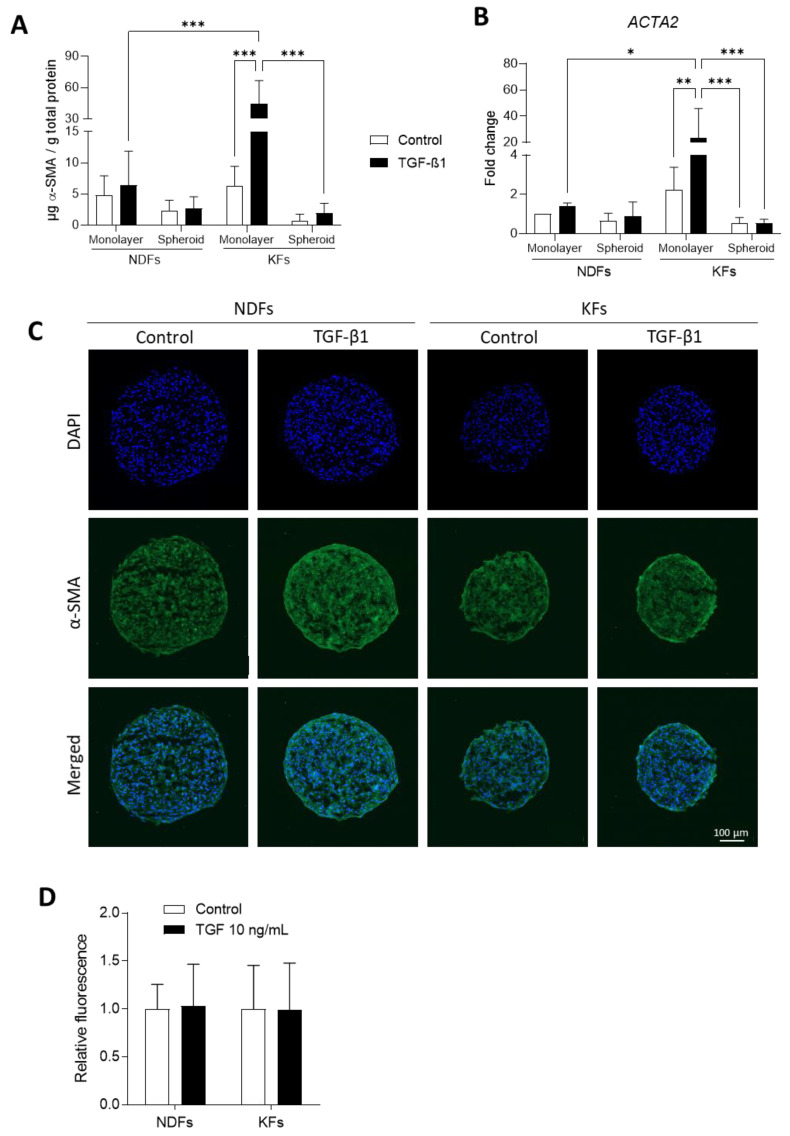
TGF-β1-induced α-SMA expression discontinues in KFs spheroids. α-SMA expression was evaluated in NDFs and KFs monolayers and spheroids after 7 days of treatment with or without TGF-β1. The expression of α-SMA was studied using ELISA (**A**) and RT-qPCR (**B**). After maturation, spheroids thus treated were immunostained for α-SMA observation (**C**) and semi quantification (**D**) [*n* = 10 spheroid and *n*= 2–6 images per spheroid]. Statistical analyses were performed using two-way ANOVA * for *p* < 0.05; ** for *p* < 0.005; *** for *p* < 0.001.

**Figure 4 biomedicines-11-02350-f004:**
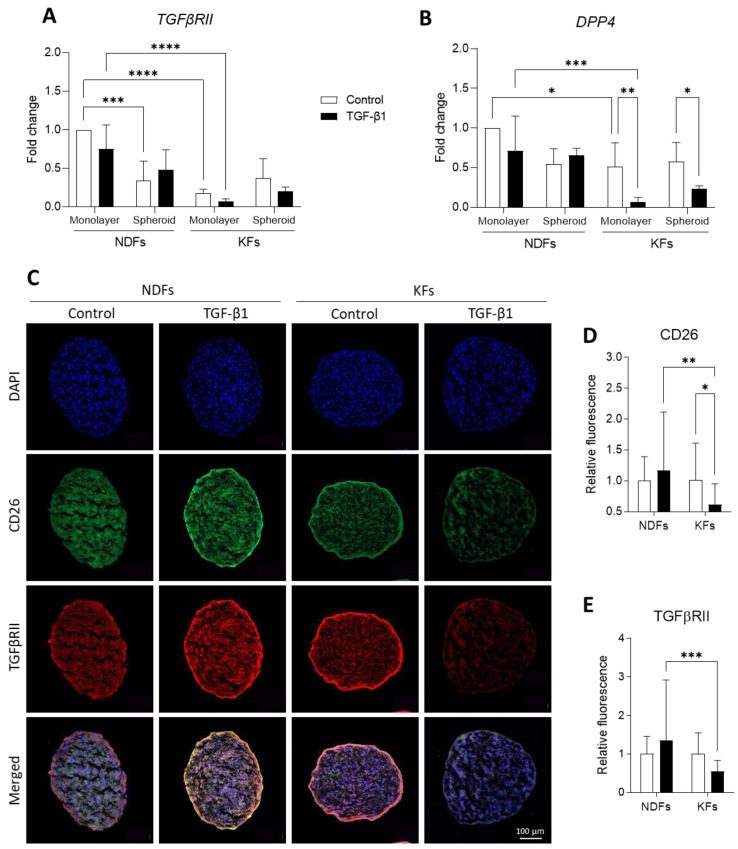
Three-Dimensional culture and TGF-β1 activation converge to downregulate CD26 and TGFβRII expression. *TGFβRII* (**A**) and *DPP4* (**B**) mRNA synthesis were evaluated by RT-qPCR in monolayers and spheroids performed either with NDFs or KFs in the presence of TGF-β1 vs. control [*n* = 4 per condition]. After 7 days of treatment, spheroids thus treated were immunostained for CD26 (green) and TGFβRII (red) observation (**C**) and semi quantification (**D**,**E**). Nuclei were counterstained with DAPI (Blue). [*n* = 10 spheroid and *n*= 2–6 images per spheroid]. Statistical analyses were performed using two-way ANOVA * for *p* < 0.05; ** for *p* < 0.005; *** for *p* < 0.001; **** for *p* < 0.0001.

**Figure 5 biomedicines-11-02350-f005:**
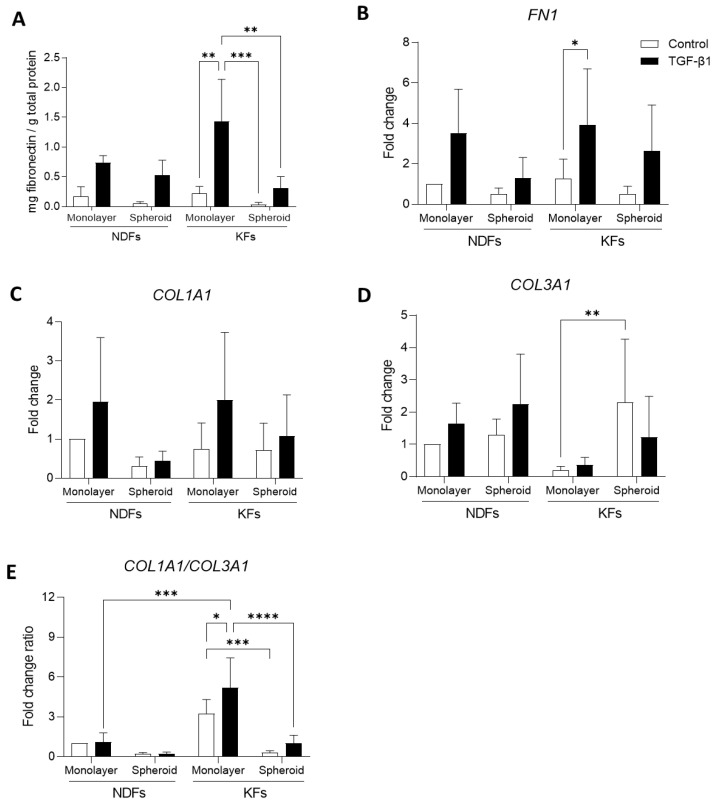
Overexpression of ECM related genes discontinues when KFs are cultured from 2D to 3D. Fibronectin expression was assessed and quantified using ELISA (**A**) and RT-qPCR (**B**) in monolayers and spheroids made with NDFs and KFs in the presence of TGF-β1 vs. control. Type I (**C**) and type III (**D**) collagen encoding mRNA, and *COL1A1*/*COL3A1* ratio (**E**) were evaluated by RT-qPCR in monolayers or spheroids produced and treated as previously described [*n* = 4 spheroids per condition]. Statistical analyses were performed using two-way ANOVA * for *p* < 0.05; ** for *p* < 0.005; *** for *p* < 0.001; **** for *p* < 0.0001.

**Table 1 biomedicines-11-02350-t001:** List of keloid donors, sex, age, and phototype.

Location	Sex	Age	Phototype
Earlobe	F	22	III
Earlobe	F	22	II
Earlobe	F	19	II
Earlobe	F	54	III

## Data Availability

The data presented in this study are available on request from the corresponding author.

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
