# Peer review of "Is Spheroid a Relevant Model to Address Fibrogenesis in Keloid Research?"

_biomedicines, 2023, doi:10.3390/biomedicines11092350_

Round 1

Reviewer 1 Report

In this manuscript, the authors examined whether 3D spheroids are useful for the assessment of keloid fibroblasts (KFs). KFs showed lower apoptosis compared to normal dermal fibroblasts (NDFs) in 3D spheroids in the presence of TGF-b1. However, fibrogenic features of KFs are downregulated in 3D spheroids compared to 2D monolayers, indicating that spheroid is not a good model for keloid fibrogenesis. This paper show the limitation of the use of spheroid for the assessment of keloid fibrogenesis.

1. Fig. 2 shows that TGF-b1 treatment induce apoptosis resistance in KFs in 3D spheroids. This phenotype may be associated with the pathogenesis of keloid. Is apoptosis resistance of KFs observed in other experimental models, such as 2D monolayers and organoids? Please discuss this point in more details.

2. The 3D spheroids used in this study were generated from isolated fibroblasts. They are different from multicellular spheroids (organoids). In the last part of Discussion, the authors discuss the involvement of the mechanical component in keloid fibroblast activation. Please discuss the difference between a 3D spheroid model and a skin organoid model and the involvement of other components, including other types of cells and extracellular matrix, in more detail.

3. It would be better if the authors could provide the improved method of spheroid with experimental evidence.

Reviewer 2 Report

Dirand et al. describes a novel 3D spheroid model of keloids, which represents a severe complication of skin burns and/or other skin defects. Indeed, treating keloid is still an uncovered clinical need that is clearly related to limited knowledge about keloid aetiology. 3D models contribute to better understanding of the disease. The manuscript is well writer with apt background information. I have some questions/feedback for the authors to improve the quality of the manuscript.

Comments:

1 Was the donor site of healthy skin biopsies matched with the scars?

2 Which passage of fibroblasts was used for 2D/spheroid culture?

3 Can you explain more about “SCAR WARS” (NCT03312166) trial?

4 In Fig 3 spheroids with KF are smaller than those generated with NDF. Is this a specific effect? Of KF?

5 The authors stated: We surprisingly observed that fibrogenic features of KFs are strongly downregulated when cells are cultured in 3D.

How this effect can by explained? What could be the reasons for fibrotic features of KF cultured in 2D? Please comment more on this interesting phenomenon.

fine

Reviewer 3 Report

In this research-based article the authors demonstrate, that, unexpectedly, keloid-derived fibroblasts (KF) lose their keloid-specific features (such as sensitivity to TGF-b1 in terms of the quantity of the downstream target proteins synthesized in response to it, levels of collagen expression, etc) if cultured in 3D spheroid culture condition. Thus, the authors demonstrate that 3D spheroids can be generated from KF and, surprisingly, such spheroids can be used as a model of the deactivation of fibrogenesis, rather then a good model of keloid pathogenesis.

Line 51. “In keloid, fibroblasts are present in high concentration compared to normal tissues [10]». Perhaps it would be better to say “in keloid, fibroblasts are present in higher numbers ...”

Line 59. “KFs cultured in 2D do not fully recapitulate the in vivo architecture, cell-cell (i.e fibroblast-immune cells) and cell-matrix interactions observed in keloid tissue”.

In such case, given the quasi-neoplastic nature of keloids, do the authors want to discuss in details the role of stem-like cells in keloid lesions, and role of keloid-associated immune cells?

Given the role of the ethnic/racial component in the keloid pathogenesis (https://www.ncbi.nlm.nih.gov/pmc/articles/PMC2884925/), what if the finding of this study is ethnic/race-specific and can not be accurately extrapolated to other race/ethnicity? What about gender-specific features of the keloid pathogenesis?

Did the authors ensured that the observed effect is not just patient-specific?

In other words, did they use KFs from several donors of different ethnicity/race/gender?

Did the author tried to use several concentrations of TGF and perhaps to increase its concentration significantly for spheroids? Maybe Kfs retain their responsiveness to TGF-b1, but in a different range?

Line 238. “Over expression” - overexpression?

Round 2

Reviewer 1 Report

All comments have been addressed easily. The authors should discuss the limitation(s) of study and future research subjects in more detail.
